# The Influence of Lablab Purpureus Growth on Nitrogen Availability and Mineral Composition Concentration in Nutrient Poor Savanna Soils

Latoya Miranda Mthimunye, Gudani Millicent Managa and Lufuno Ethel Nemadodzi *

Department of Agriculture and Animal Health, University of South Africa,
Johannesburg 1709, Florida, South Africa
* Correspondence: nemadle@unisa.ac.za; Tel.: +27-11-670-9586

**Abstract:** Low soil fertility in savanna soils has been linked to low crop yields, with nitrogen being the most limiting factor in crop yield. Soil used in this pot experiment was obtained from Motshephiri village with low total N, low $NO_3^-$ and high $NH_4^+$. A pot experiment was conducted in a greenhouse laid in a Randomized Complete Block Design with four treatments (1) control, (2) *Bradyrhizobium japonicum* inoculant, (3) superphosphate and (4) *Bradyrhizobium japonicum* inoculant + superphosphate). The superphosphate was applied at three different levels (45, 60 and 75 kg/ha). Lablab was cultivated in each treatment and the results of the study indicated that lablab growth significantly increased total N and $NO_3^-$, and reduced concentration $NH_4^+$ relative to the original soil herein referred to as pre-lablab growth. However, the N forms (total N, $NO_3^-$ and $NH_4^+$) did not differ significantly amongst different levels of superphosphate with or without *Bradyrhizobium japonicum* inoculant. Lablab growth, proved to have a significant impact on both the soil macro (P, K, Ca, Mg, and Na) and micronutrient level (Fe, Mn, Cu, B and Cl) with the exception of Zn. This study suggests that lablab's ability to rapidly boost soil N content, overall soil fertility in a short period of time without the use of superphosphate fertilizers or *Bradyrhizobium japonicum* inoculants makes it ideal for intercropping or rotating with non-leguminous crops that have a short growing season.

**Keywords:** nitrogen fixation; nitrogen forms; total nitrogen; legumes; soil nutrient status





## 1. Introduction

Savanna soils occupy about 20% of the earth's surface and 54% of the Southern African area [1,2] providing ecosystem services such as livestock feed, soil carbon sequestration, fiber production, soil conservation, and recreation [3,4]. Savanna ecosystems have been characterized by the presence of sub-shrubs, shrubs, trees and grasses, in different proportions and temporal patterns associated to seasonality and fire [5]. The occurrence of savannas and their physiognomic gradient have been reported to be maintained by fires deriving from lightning strikes and anthropogenic sources vital to balance the vegetation distribution, diversity and to clear the land for agricultural practices [6,7]. Consequently, a major proportion of nitrogen (N) in the litter burned is volatilized and thus lost from the soil. Ref. [1] postulated that it is for this reason savanna soils are often inadequate of N. In addition to the loss of N due to burning litter, in many parts of Africa, savanna soils are threatened by declining soil fertility owing to the continuous land exploitation in response to increasing human population pressures [8]. Studies indicated that conversion of savanna lands to croplands is the main driver of soil nutrient dynamics [9,10]. Additionally, the low soil fertility levels in savanna lands have been found to contribute to the low crop yield with N being the main limiting factor of growth in crops [10].

Nitrogen is a nutrient element incorporated in the plant tissue making about 1 to 5% of the dry matter [11]. It is a primary essential component of the chlorophyll molecule necessary for photosynthesis, is vital for plant growth and development, plants need N in

large quantities to generate amino acids that form part of nucleic acids and proteins [12]. Plants that receive enough N grow rapidly and produce large green foliage, although it is known that not all the N in the soil is accessible for plant uptake [13]. Plants obtain their N from the soil in a form of nitrate ($NO_3^-$) and ammonium ($NH_4^+$) ions which only make up a small proportion in the soil, less than five percent, with the rest being the soil organic N which is gradually transformed to inorganic N through the process of mineralization [13]. Ref. [12] reported that, of these forms of N accessible to plants, $NO_3^-$ is the major form of N absorbed by plants and it is more mobile in the soil than $NH_4^+$. Conversely, $NO_3^-$ is susceptible to leaching since it possesses a negative charge which causes repulsion to the negative charges on the soil particles [14]. In soils deficient of N, $NH_4^+$ content tends to be greater than $NO_3^-$ due to the positive charge of $NH_4^+$ that attracts the negatively charged soil particles [15,16]. Ammonium that is not absorbed by the plants is prone to nitrification where it is converted to $NO_3^-$ by microorganisms [13]. Among other sources of soil N namely, manure, crop residues, and chemical fertilizers, legumes also serve as soil N source that are reliable in replenishing low soil nutrient levels [16].

Lablab is drought tolerant that enables its foliage to remain green throughout the dry season when other legumes such as common bean and cowpea have dried out, which ultimately makes it a good forage supplement for livestock [17,18]. In addition, Lablab, unlike other summer legumes such as velvet bean, can establish well under various environmental conditions as it can withstand temperature as low as 3 °C [19,20].

The study focused on Lablab (*Lablab purpureus*), a forage legume in the Fabaceae family, which includes among other legumes; cowpea, pigeon pea, and soybean [17,18]. Lablab was specifically chosen for the current study because it is widely consumed in the Northern province of South Africa, easily accessible and grown by small holder farmers in the province. Lablab is an African native crop that grows in tropical and subtropical climates [21], an annual to short-lived perennial crop which grows well in summer. This multi-purpose leguminous crop grows well in heavy soils but has also been reported to grow in sandy soils with the pH of 5 to 7.5 [22] and has a high water-use efficiency and drought tolerant and can remain green throughout the dry season when other fodder crops are dry and scarce [23,24]. Studies revealed that lablab performs well when incorporated in intercropping farming systems, particularly with maize, in relation to sufficient forage and grain yield, and N fixation [25,26]. Moreover, ref. [21] reported that lablab can also be intercropped with sorghum and pearl millet due to its indeterminate growth and that lablab is effective in protecting the soil against erosion [27].

The commonly known lablab cultivars are Highworth which are the early flowering and the Rongai, the late flowering cultivar. The two above-mentioned cultivars are similar in morphology. Highworth cultivar is characterized by light brown seeds, white flowers and is generally preferred for intercropping since it is a less robust climber than Rongai. On the contrary, Rongai is characterized by black seeds and purple flowers [22]. Although lablab is not widely used in South Africa, its global popularity is demonstrated and supported by a different common name documented by various authors and databases. According to data compiled by [19], lablab common names among others include Hyacinth bean (Fiji, Naura, Philipines and Java), Country bean (Bangladesh and Gazipur), Lubia bean (Ethiopia). In South Africa, it is referred to as Mabonjisi particularly in the Limpopo province and is widely consumed by VhaVenda, VaTsonga and BaPedi tribes. A survey with smallholder farmers in rural areas in Limpopo province revealed that, lablab leaves (dried or fresh) may be cooked and eaten as leafy vegetables [28].

Lablab crop is grown in several parts of the world due to its multi-purpose potential; in the western part of Africa, such as Malawi and Ethiopia, it is utilized as a vegetable and consumed as boiled immature seeds [19]. In Australia, is cultivated as a forage crop for livestock feed as well as green manure to improve soil productivity [29]. It is reported as the daily supplier of protein consumed as processed food in India [30]. Moreover, studies have shown lablab's medicinal benefits which serve as nutraceuticals and pharmaceuticals in treating various diseases such as hyperlipidemia, osteoporosis, and

pancreatic cancer [31,32]. Globally, legumes have long been recognized for their ability to fix N into the soil, thus improving soil fertility. They have been and are estimated to fix up to 200 kg $N_2$ per hectare provided that the soil is well-drained with a temperature of not less than 7 °C [24,29,33]. For legume to fix N efficiently, seeds must be inoculated with the relevant strain of rhizobia to influence and /or encourage nodulation [34]. Additionally, [35] reported that inoculating lablab seeds with *Bradyrhizobium japonicum* strain was found to have been effective on nodulation and nitrogen fixation capacity of lablab.

Phosphorus is generally present as insoluble mineral phosphate in the soil being prone to fixation which becomes unavailable for plant uptake [36]. A significant amount of phosphorus is required for nodulation which is vital for nitrogen fixation in legumes plants [37]. However, legumes may experience limited nitrogen fixation when they do not receive enough phosphorus, which may lead to nitrogen deficiency [38]. A study [39] reported that lablab reached the highest nutrients uptake at a dose of 60 kg/ha of phosphorus fertilizer.

The current study was thus formulated to explore lablab performance as a strategic approach to improve soil fertility and encourage organic farming by including lablab since it possesses other remarkable potential uses that farmer can benefit from. Three levels of superphosphate were applied, 60 kg/ha as adopted from [39], 45 kg/ha and 75 kg/ha which were below and above the recommended 60 kg/ha in order to further determine the lablab performance when cultivated in Savanna soils.

In the past years, agricultural practices have been field-based, however, due to the increase in the number of human population which is slowly reducing the arable land for cultivation and/or cropping systems. Developing countries are diverting into hydroponics and controlled environments such as shade nets, glasshouses, greenhouses to mitigate the insufficient land space for crop production [40–42]. Although the adoption rate of these agricultural strategies is low in developing countries, including South Africa, they are viewed as the vital as the advance method of farming and a means to eradicate poverty [43] which will ultimately be the new ways of farming. In rural areas of South Africa, the high cost of chemical fertilizers prohibits fertilizer utilization among subsistence farmers. Although, chemical fertilizers enhance rapid growth, they are not environmentally friendly. Therefore, the objectives of the study were to (1) to evaluate the impact of lablab growth on soil nitrogen content of savanna soil, (2) to determine the effect of lablab growth on the chemical properties if savanna soils.

## 2. Materials and Methods

### Soil sampling area and sampling strategy

The soil used in the study was collected from Motsephiri Village, Germsbokspruit, under Elias Motsoaledi local municipality in Sekhukhune district of Limpopo province, South Africa (latitude: 25°0′59.6″ longitude: 29°42′53″). The area has a semi-arid climate characterized by cold dry winters and warm rainy summers [44–46]. It receives a mean annual rainfall of 604 mm, with an annual temperature which ranges from 8 to 28 °C [45,46], at an altitude of 1525 m above sea level. It is dominantly characterised by sandy clay loam soil textural class dominant and is known to be used for monocropping of maize.

In September 2021, five 10 m × 10 m replicate plots were randomly demarcated in the field. Soil samples were collected using a zigzag procedure of each replicate plot using a spade at the depth of 30 cm. The samples from each plot were then thoroughly mixed into a 25-L bucket, yielding five composite samples. The composite soil samples were then transferred into labeled sacks and transported to the University of South Africa greenhouse at Florida Science Campus where they were sieved through a 4 mm metal sieve to remove roots and rocks. The samples were divided into two sub samples which were used for soil physical and chemical analysis (pre-planting) and pot planting experiment.

### Pre-planting soil physical and chemical analysis

Soil was subjected to physical and chemical analysis which was done at the Agricultural Research Council Institute for Soil, Climate, and Water (ARC-ISCW) situated in Arcadia, Pretoria. Each sub-composite soil sample was analyzed for soil particle distribu-

tion as prescribed by [47], soil pH [48], organic carbon [49], total nitrogen [50], mineral nitrogen [51], extractable phosphorus [52], soil macronutrients [53] and soil micronutrients (chlorine and boron) were measured by 1:10 water extract method [54] while zinc, copper, iron, and manganese were measured by 0.1 HCl extract method [55].

**Post-planting analysis**

After the termination of the pot experiment, the root systems of the lablab were excavated from the pots and shaken to detach the lablab rhizosphere soil, in preparation for chemical soil analysis post-lablab growth. To compare the differences in soil mineral elements pre- and post-lablab growth, the soil was air-dried and sieved through a 2 mm [56] sieve to remove root debris.

**Experimental design and layout**

The pot experiment was conducted in the greenhouse with a minimum and maximum air temperature ranges of 7.4 and 44.9 °C, which is situated at the University of South Africa, Florida Science Campus, Rooderpoot (Latitude: $-26°9'29.274''$; Longitude: $27°55'17.663''$). The average relative humidity inside the greenhouse was 68% during the planting period (October 2021 to February 2022). The plastic pots (18 cm diameter, 14.5 cm height and 18 cm width) were laid out in a Randomized Complete Block Design (RCBD) with four treatments. Each treatment at each level of single superphosphate was replicated four times under five composite samples totaling to 160 pot plants. Lablab seeds of early maturing Highworth cultivar was used in the experiment. The treatment consisted of a control, *Bradyrhizobium japonicum* inoculant, single superphosphate (8.3% P) applied at 45, 60 and 75 kg/ha and *Bradyrhizobium japonicum* inoculant plus single superphosphate applied at 45, 60 and 75 kg/ha.

Single superphosphate granules were placed 2 cm below the seeds at the time of sowing at a rate of 0, 45, 60 and 75 kg/ha in relevant treatments. The seeds were evenly coated with *Bradyrhizobium japonicum* inoculant applied at the rate of 50 kg/200 mL. 2030 g of soil was used to fill each pot, with two lablab seeds planted at a depth of 3 cm. The plants were irrigated with 250 mL of tap water every second day. Two weeks after sowing, the plants were thinned to one plant per pot, weeding was done manually as and when the need arise, and insects were controlled by means of a non-systematic insecticide (organophosphate). The experiment ran for a period of 6 months, with lablab grown in each treatment replicated four times. The weather and/or temperature was automatically controlled and regulated to fit the weather of Roodepoort, Johannesburg where the greenhouse is situated.

**Statistical analysis**

The significant differences ($p < 0.05$) in soil N content pre- and post-lablab planting were determined using a one-way analysis of variance (ANOVA) with the 'aov' function in the agricolae R package [56]. Mean separations among treatments (i.e., pre-lablab, control, *Bradyrhizobium japonicum* inoculant, different levels of single superphosphate as well as *Bradyrhizobium japonicum* inoculant + different single superphosphate levels on lablab plants) were performed with one way ANOVA using Duncan's multiple range test at 5% probability.

## 3. Results

### 3.1. The Effect of Lablab Growth on Soil Nitrogen Forms

3.1.1. Effect of Lablab Growth on Soil Total Nitrogen

The content of total N in savanna soil was significantly higher in the soil planted with lablab (control) as compared to the initial bulk soil analyzed pre-lablab growth (Table 1). However, the total N did not differ significantly between the applied *Bradyrhizobium japonicum* inoculant and/or superphosphate treatments. The significant increase of total N content in control where lablab was planted without amendment by *Bradyrhizobium japonicum* inoculant and/or superphosphate relative to soil pre-lablab accounted 0.094% as shown in Table 1.

**Table 1.** Comparison of nitrogen source on savanna soil pre-lablab versus post-lablab growth.

| | Total N (%) | $NO_3^-$ (mg/kg) | $NH_4^+$ (mg/kg) |
|---|---|---|---|
| Pre-lablab | 0.083 [b] | 3.67 [b] | 40.8 [a] |
| Post-lablab | | | |
| T1 (Control) | 0.094 [a] | 9 [a] | 12.20 [b] |
| T2 SP45 (45 kg/ha Superphosphate) | 0.092 [ab] | 7.6 [a] | 15.2 [b] |
| T3 SP60 (60 kg/ha Superphosphate) | 0.087 [ab] | 6.97 [a] | 11.7 [b] |
| T4 SP75 (75 kg/ha Superphosphate) | 0.088 [ab] | 7.55 [a] | 18.1 [b] |
| T5 (*Bradyrhizobium japonicum* inoculant) | 0.086 [ab] | 3.53 [b] | 15.3 [b] |
| T6 B + SP45 (*Bradyrhizobium japonicum* + 45 kg/ha of Superphosphate) | 0.091 [ab] | 8.35 [a] | 14.3 [b] |
| T7 B + SP60 (*Bradyrhizobium japonicum* + 60 kg/ha of Superphosphate) | 0.088 [ab] | 7.05 [a] | 12.9 [b] |
| T8 B + SP75 (*Bradyrhizobium japonicum* + 75 kg/ha of Superphosphate) | 0.088 [ab] | 7.76 [a] | 17.7 [b] |
| s.e.d. | 0.0042 | 1.163 | 4.46 |
| LSD (5%) | 0.0086 | 2.359 | 9.05 |

The letters [a], [b] in the same column denotes statistical differences between means as predicted using the Duncan's multiple range test at 5% probability level.

### 3.1.2. Effect of Lablab Growth on Soil Nitrate Content

The growth of lablab had a significant positive effect on the soil nitrate ($NO_3^-$) levels in all the treatments with the exception of *Bradyrhizobium japonicum* inoculant treatment, here in referred as T2, which did not differ significantly with the pre-lablab soil. Moreover, *Bradyrhizobium japonicum* inoculant and phosphorus supplementation at all levels did not show significant influence on the soil $NO_3^-$ content.

### 3.1.3. Effect of Lablab Growth on Soil Ammonium Content

Ammonium ($NH_4^+$) levels in the soil post-lablab growth declined significantly in all the treatments relative to the pre-lablab soil (40.8 mg/kg). Nevertheless, the decrease in the $NH_4^+$ level was not significant amongst the *Bradyrhizobium japonicum* inoculant, superphosphate and *Bradyrhizobium japonicum* + superphosphate treatments The mean values of the treatments ranged from 12.2 mg/kg recorded in control to 18.1 mg/kg recorded in 75 kg/ha superphosphate treatment as indicated in Table 1.

### 3.2. The Effect of Lablab Growth on Soil Macronutrients
#### 3.2.1. Phosphorus

The soil analysis results showed that a slight increase "no significant" difference occurred regarding phosphorus (P) content when lablab was planted without treatment (17.31 mg/kg) or treated with *Bradyrhizobium japonicum* inoculant (15.38 mg/kg) as compared to pre-lablab (15.87 mg/kg). However, notably, the incorporation of superphosphate solely or together with *Bradyrhizobium japonicum* inoculant, increased P significantly. Furthermore, P increased with the increased in superphosphate level with or without *Bradyrhizobium japonicum* inoculant as indicated in Table 2.

#### 3.2.2. Potassium

Exchangeable potassium (K) content was recorded the highest (200.6 cmol (+)/kg) where lablab was planted without amendments by superphosphate and/or *Bradyrhizobium japonicum* inoculant as compared to the pre-lablab (0.39 cmol (+)/kg) which accounted a significant difference of 200.21 cmol (+)/kg as shown in Table 2. All treatments resulted in significant increase in K content relative to pre-lablab. The rate of application of superphosphate at different levels did not change K content significantly, however, a significance

increase was observed between 45 and 60 kg/ha superphosphate plus *Bradyrhizobium japonicum* inoculant treatments.

**Table 2.** Comparison of macronutrient content, cation exchange capacity, and soil pH of savanna soil pre- versus post- lablab growth.

| Macronutrient Content | P mg/kg | K cmol(+)/kg | Ca cmol(+)/kg | Mg cmol(+)/kg | Na cmol(+)/kg | CEC cmol(+)/kg | pH (H$_2$O) | pH (KCL) | OC (%) |
|---|---|---|---|---|---|---|---|---|---|
| **Pre-lablab** | 15.87 [d] | 0.39 [c] | 1.15 [b] | 0.51 [b] | 0.04 [c] | 9.45 [ab] | 5.13 [c] | 4.27 [b] | 1.41 [a] |
| **Post lablab** | | | | | | | | | |
| T T1 (Control) | 17.31 [cd] | 200.6 [a] | 332.6 [a] | 104.68 [a] | 38.1 [ab] | 9.54 [ab] | 5.65 [ab] | 4.86 [a] | 1.43 [a] |
| TT2 SP45 (45 kg/ha Superphosphate) | 29.86 [abc] | 181.4 [ab] | 298.6 [a] | 97.92 [a] | 48.3 [a] | 11.77 [a] | 5.49 [b] | 4.66 [a] | 1.44 [a] |
| TT3 SP60 (60 kg/ha Superphosphate) | 32.9 [ab] | 151 [b] | 291 [a] | 90.28 [a] | 35.86 [ab] | 8.22 [ab] | 5.54 [ab] | 4.68 [a] | 1.39 [a] |
| T T4 SP75 (75 kg/ha Superphosphate) | 41.87 [a] | 176.2 [ab] | 332.8 [a] | 101.82 [a] | 45.34 [ab] | 8.1 [ab] | 5.56 [ab] | 4.75 [a] | 1.45 [a] |
| TT5 (*Bradyrhizobium japonicum* inoculant) | 15.38 [d] | 191.4 [a] | 317 [a] | 103.58 [a] | 33.5 [b] | 8.92 [ab] | 5.81 [a] | 4.98 [a] | 1.41 [a] |
| TT6 B + SP45 (*Bradyrhizobium japonicum* + 45 kg/ha of Superphosphate) | 26.3 [cd] | 156 [b] | 292.6 [a] | 96.78 [a] | 40.42 [ab] | 6.07 [b] | 5.53 [ab] | 4.72 [a] | 1.38 [a] |
| T T7 B + SP60 (*Bradyrhizobium japonicum* + 60 kg/ha of Superphosphate) | 34.67 [ab] | 192.4 [a] | 312.6 [a] | 102.2 [a] | 44.7 [ab] | 7.33 [ab] | 5.56 [ab] | 4.79 [a] | 1.48 [a] |
| TT8 B + SP75 (*Bradyrhizobium japonicum* + 75 kg/ha of Superphosphate) | 41 [a] | 184.4 [ab] | 328.4 [a] | 106.26 [a] | 36 [ab] | 7.73 [ab] | 5.57 [bc] | 4.75 [a] | 1.44 [a] |
| s.e.d. | 6.16 | 16.21 | 29.21 | 8.96 | 7.76 | 2.233 | 0.1289 | 0.139 | 0.0503 |
| LSD (5%) | 12.49 | 32.89 | 59.25 | 18.17 | 15.73 | 4.528 | 0.2614 | 0.287 | 0.1020 |

The letters [a], [b], [c] and [d] in the same column denotes statistical differences between means as predicted using the Duncan's multiple range test at 5% probability level.

### 3.2.3. Calcium

Calcium (Ca) content in the soil increased significantly in all the treatments as compared to recorded in pre-lablab (* see Table 2). Lablab growth increased Ca content significantly regardless of supplementation of the soil by superphosphate or treating lablab seeds with *Bradyrhizobium japonicum* inoculant. Although the amendment by superphosphate and/or *Bradyrhizobium japonicum* inoculant resulted in a significant increase in Ca content, applying superphosphate at different levels did not yield any significant change. The significant difference between the highest value recorded in control where lablab was planted alone (332.6 cmol (+)/kg) and the lowest value recorded in pre-lablab (1.15 cmol (+)/kg) was 331.45 cmol (+)/kg which accounted for 99.31% significant increase.

### 3.2.4. Magnesium

Similarly, magnesium (Mg) soil content showed a significantly increase in all the treatments above that recorded in pre-lablab. Lablab growth had a positive influence in Mg content regardless of supplementation of the soil by superphosphate or treating lablab seeds with *Bradyrhizobium japonicum* inoculant. Mg content did not vary significantly at different levels of superphosphate whether applied solely or with *Bradyrhizobium japonicum* inoculant. The highest Mg content was recorded in *Bradyrhizobium japonicum* + 75 kg/ha of superphosphate (106.26 cmol (+)/kg) and the lowest value in the pre-lablab (0.51 cmol (+)/kg) as indicated in Tables 2 and 3.

### 3.2.5. Sodium

Lablab growth increased sodium (Na) soil content in all treatments compared with the pre-lablab whether amended with superphosphate and/or *Bradyrhizobium japonicum* inoculant. The highest Na concentration was recorded in 45 kg/ha superphosphate (48.3 cmol (+)/kg) and the lowest in pre-lablab (0.035 cmol (+)/kg).

**Table 3.** Comparison of micronutrient content of savanna soil pre- versus post-lablab growth.

| Micronutrient Content | Fe (mg/kg) | Mn (mg/kg) | Cu (mg/k) | Zn (mg/k) | B (mg/kg) | Cl (mg/kg) |
|---|---|---|---|---|---|---|
| **Pre-lablab** | 0.3 [b] | 31.1 [b] | 1.05 [b] | 2.53 [a] | 0.19 [b] | 5.7 [b] |
| **Post lablab** | | | | | | |
| T1 (Control) | 21.3 [a] | 45.6 [ab] | 1.23 [a] | 2.91 [a] | 0.704 [a] | 28.4 [a] |
| T2 SP45 (45 kg/ha Superphosphate) | 14.92 [a] | 48.2 [ab] | 1.15 [ab] | 3.03 [a] | 0.818 [a] | 36.4 [a] |
| T3 SP60 (60 kg/ha Superphosphate) | 15.6 [a] | 51.2 [a] | 1.14 [ab] | 2.87 [a] | 0.744 [a] | 31.9 [a] |
| T4 SP75 (75 kg/ha Superphosphate) | 15.5 [a] | 50.2 [a] | 1.15 [ab] | 2.9 [a] | 0.692 [a] | 32.6 [a] |
| T5 (*Bradyrhizobium japonicum* inoculant) | 16.36 [a] | 55 [a] | 1.23 [a] | 3.3 [a] | 0.744 [a] | 27.4 [a] |
| T6 B + SP45 (*Bradyrhizobium japonicum* + 45 kg/ha of Superphosphate) | 14.14 [a] | 46.7 [ab] | 1.11 [ab] | 2.62 [a] | 0.628 [a] | 30.8 [a] |
| T7 B + SP60 (*Bradyrhizobium japonicum* + 60 kg/ha of Superphosphate) | 15 [a] | 49.5 [ab] | 1.10 [ab] | 3.51 [a] | 0.842 [a] | 3.5 [a] |
| T8 B + SP75 (*Bradyrhizobium japonicum* + 75 kg/ha of Superphosphate) | 15.16 [a] | 49.5 [ab] | 1.21 [ab] | 3.01 [a] | 0.642 [a] | 32.8 [a] |
| s.e.d. | 3.432 | 8.18 | 0.0714 | 0.1475 | 0.1028 | 4.29 |
| LSD (5%) | 6.961 | 16.59 | 0.1448 | 0.964 | 0.2085 | 8.69 |

The letters [a], [b] in the same column denotes statistical differences between means as predicted using the Duncan's multiple range.

### 3.3. Cation Exchange Capacity

Lablab growth did not have a significant influence on the soil cation exchange capacity (CEC). It is however worth noting that there was a slight decrease of soil CEC in *Bradyrhizobium japonicum* inoculant, 60 and 75 kg/ha superphosphate and *Bradyrhizobium japonicum* + superphosphate applied 45, 60 and 75 kg/ha treatments with mean values of 8.92, 8.22, 8.1, 6.07, 7.33 and 7.73 cmol (+)/kg respectively and a slight increase in control (9.54 cmol (+)/kg) and 45 kg/ha superphosphate treatment (11.77 cmol (+)/kg) as compared to pre-lablab (9.45 cmol (+)/kg).

### 3.4. Soil pH

Lablab growth resulted in a significant increase in the soil pH with a mean range of 5.49 to 5.81 for pH across the treatments compared to pre-lablab was 5.13 (refer to Table 2). Moreover, there was no significant change observed and recorded at superphosphate levels solely applied or with *Bradyrhizobium japonicum* inoculant in the soil pH. There was however a significant difference observed between *Bradyrhizobium japonicum* inoculant and 45 kg/ha superphosphate treatments in terms of soil pH.

### 3.5. The Effect of Lablab Growth on Soil Micronutrients
3.5.1. Iron

The soil iron (Fe) content was significantly influenced by lablab growth. The results show a significant increase in extractable soil Fe in all the treatments with a mean range of 14.14 to 21.3 mg/kg as opposed to the pre lablab soil with a mean of 0.3 mg/kg. However, the significant increases following lablab growth did not vary amongst the different levels of superphosphate (45, 60 and 75 kg/ha superphosphate) when applied alone or with *Bradyrhizobium japonicum* inoculant as indicated in Table 3.

3.5.2. Manganese

Lablab growth resulted in a significant increase in the soil manganese (Mn) content observed in *Bradyrhizobium japonicum* inoculant and superphosphate applied at 60 and 75 kg/ha treatments with mean values of 55, 50 and 51 mg/kg respectively as compared to the pre lablab soil with a mean value of 31.1 mg/kg. The Mn soil content did not vary significantly at different levels of superphosphate. Moreover, the interaction of *Bradyrhizobium*

*japonicum* inoculant and superphosphate fertilizer did not have any influence on the soil extractable Mn.

### 3.5.3. Copper

In comparison to the pre lablab soil with a mean of 1.05 mg/kg, lablab growth without superphosphate fertilizer and/or inoculant significantly increased the content of extractable copper (Cu) in the soil (1.23 mg/kg). Treating lablab seeds with the *Bradyrhizobium japonicum* inoculant pre growth resulted in positive influence in soil Cu content. On the contrary, when *Bradyrhizobium japonicum* inoculant was applied together with the superphosphate, the Cu did not increase significantly in the soil. In the same way, Cu content did not vary at different levels of superphosphate (45, 60 and 75 kg/ha).

### 3.5.4. Zinc

The findings of the study indicates that the growth of lablab alone or supplemented with superphosphate and/or *Bradyrhizobium japonicum* inoculant did not have any effect on zinc concentration in the soil with the mean range of 2.53 to 2.91 mg/kg as shown in Table 3 below.

### 3.5.5. Boron

All the treatments increased boron (B) content significantly compared to the pre-lablab (Table 3, with the highest mean value (0.842 mg/kg) recorded in T4 B + SP60 *Bradyrhizobium japonicum* + 60 kg/ha of superphosphate and the lowest value (0.19 mg/kg) recorde in the pre-lablab. Applying superphosphate at different levels did not influence B content significantly (refer to Table 3).

### 3.5.6. Chloride

The same trend was observed in chloride (Cl) as it was in Fe and B as shown in Table 3 below, lablab growth increased Cl significantly across the treatments, however, the significance increase did not vary significantly at different levels of superphosphate whether applied solely or with *Bradyrhizobium japonicum* inoculant with the exception of T4 B + SP60 *(Bradyrhizobium japonicum* + 60 kg/ha of superphosphate treatment which did not differ significantly as compared to the pre-lablab. The lowest soil Cl content (3.5 mg/kg) was observed in *Bradyrhizobium japonicum* + 60 kg/ha of superphosphate treatment whiles the highest value (36.4 mg/kg) was observed when superphosphate was applied at 45 kg/ha.

## 4. Discussion

### 4.1. The Effect of Lablab Growth on Soil Total Nitrogen Content

In the Pre-lablab growth soil, the total nitrogen (N) was low (0.083%) according to [57], who reported that soil total N requirement is low at the value of less than 0.1%, medium at 0.1–0.2% and high when total N is greater than 0.2%. Based on the history of the sampling site (Motshepiri village), the low soil total N might have been accounted to continuous cultivation of maize crop with low inputs of manure or other organic fertilizers. From the results of this study, it is therefore important that farmers avoid nutrients depletion which may be caused by long periods of monocropping by practicing crop rotation with legume crops such as lablab, groundnuts, cowpea etc. In this study, the total N content of savanna soil increased significantly in the lablab rhizosphere (0.094%) compared to the soil pre-lablab growth (0.083%). Several studies reported similar results in different legume species, for instance, [58], reported that about 64% of N in the soil was released from peas (*Pisum sativum* L.) and oats (*Avena sativa* L.) roots in a form of root exudates and degraded root cells in field trials. [59] revealed that cowpea (*Vigna unguiculata*) contributed 52% of N through rhizodeposition under field condition. The N deposited by legume roots and nodules is in a form of low-weight molecular compounds (e.g., amino acids) as reported by [60], who conducted a study on white clover (*Trifolium repens*) rhizodeposition and N

dynamics. Therefore, the potential of legume species to exudate organic N compounds into the rhizosphere might be attributed to the increase in total N in the present experiment, since total N is composed of organic forms of N such as amino acids (serine, glycine, lutamine, glutamate and aspartate) as reported by [61]. Furthermore, the increase in total N combined with a considerable drop in ammonium ($NH_4^+$) (produced by mineralization) in the current study suggests that the root-derived organic N resisted mineralization for the duration of the pot experiment and constituted the total N thereof.

### 4.2. The Effect of Lablab Growth on Nitrogen Forms

The findings of the study showed that soil N is strongly influenced by root-soil interactions. Following lablab growth, nitrate ($NO_3^-$) concentration increased significantly (9 mg/kg) in lablab-planted soil despite being amended by phosphorus (P) fertilizer or *Bradyrhizobium japonicum* inoculant relative to the soil pre-lablab growth (3.67 mg/kg). In contrast, the presence of lablab resulted in significant decline in ammonium ($NH_4^+$) by the end of the pot experiment. The increase in $NO_3^-$ concentration and decrease in $NH_4^+$ in the soil post lablab growth is suggested to be due to the activity of ammonia-oxidizing microbes which converts ammonium to nitrate [62].

The results from the current study are in accordance with the findings of [63], which revealed that elevated $NO_3^-$ concentrations in the rhizosphere of all treatments at all growth stages with a maximum mean value of 180 mg/kg recorded at the early plant development stage. The same study reported that levels of $NH_4^+$ were lower in the rhizosphere of alfalfa (*Medicago sativa* L.) planted in a pot experiment in a greenhouse with a mean value of 13 mg/kg as compared to the control and did not significantly respond to inoculant treatment. Another study revealed that cereals and grasses grown from pot experiments showed a statistical decrease in $NH_4^+$ (18 mg/kg) and increase in $NO_3^-$ (80 mg/kg) concentration in soil planted with *Populus fremontii* as compared to the initial soil with 31 and 40 mg/kg of $NH_4^+$ and $NO_3^-$ respectively [64].

Since the sampling site has been continuously planted with maize over the last decade, the low initial soil $NO_3^-$ concentration was likely caused by the higher uptake of $NO_3^-$ by maize over $NH_4^+$ which could have resulted to the depletion of $NO_3^-$ in the soil [65]. Alternatively, the high initial soil $NH_4^+$ concentration could possibly be due to $NH_4^+$ fixation in the soil which hinders $NH_4^+$ uptake despite being present in the soil [66]. The growth of lablab has shown to be effective in this regard since it significantly increased $NO_3^-$ content where lablab was grown solely as shown in Table 2 which will be available for the following crop (i.e., maize) which possibly prefers it [67]. In disagreement with the present findings, studies on legumes reported that most legumes prefer $NO_3^-$ as their form of N instead of $NH_4^+$ and this means that they are likely to deplete the $NO_3^-$ in the soil and thus resulting in $NH_4^+$ accumulation [68,69]. Interestingly, $NH_4^+$ showed to have declined significantly in all the treatments, whether lablab was planted alone or supplemented with superphosphate and/or *Bradyrhizobium japonicum* as opposed to pre-lablab, which could be attributed to the assimilation of $NH_4^+$ by lablab or microbes. Lablab growth have proven to be effective in N cycling, it's decreasing effect on $NH_4^+$ is vital in preventing possible $NH_4^+$ toxicity to the $NH_4^+$ sensitive agricultural plants such as barley which was found to perform poorly in growth when $NH_4^+$ was applied at 10 mmol/L [70]. Moreover, the decreasing effect of lablab on $NH_4^+$ may indicate the potential of lablab to release the previously fixed $NH_4^+$ which was inaccessible to plant uptake because plant roots can affect $NH_4^+$ pool indirectly by exporting roots exudates that induce microbial activity and N assimilation [71].

### 4.3. The Effect of Lablab Growth on Soil Macronutrients

The results of the P soil test post lablab growth showed that the soil extractable P level did not change considerably in the control (17.31 mg/kg) and *Bradyrhizobium japonicum* inoculant (15.38 mg/kg) treatments where no superphosphate was supplied as opposed to the pre-lablab growth (15.87 mg/kg). In agreement with the current study, the findings of

the pot experiment conducted by [72], revealed that there was non-significant difference in terms of extractable P between unplanted soil (20.6 mg/kg) and soil planted with *Phaseolus vulgaris* (19.8 mg/kg). Nonetheless, the incorporation of superphosphate into the soil solely or in combination with inoculant applied on lablab seeds increased the soil extractable P significantly which followed the order of 45 kg/ha < 60 kg/ha < 75 kg/ha. The increase in the soil extractable P can be attributed to the contribution of labile P from superphosphate application and the soil pH which is considered favourable for P availability as stated by [73], who consider optimum pH range for nutrient availability to be 5.4 to 6.1. The increase in soil extractable P following superphosphate application in a legume-planted soil is consistent with a previous study that found a significant increase in extractable P from 7.34 to 23.52 mg/kg in a cowpea-planted soil with moderate acidity (pH = 5.86) following superphosphate application under field conditions which was also conducted for three months [74].

The application of superphosphate and/or *Bradyrhizobium japonicum* inoculant had a great impact on lablab's potential to influence soil exchangeable bases in all the treatments. Single superphosphate is reported to contain high quantities of calcium (Ca) which could possibility attribute to the elevated soil Ca following the amendment of the soil by superphosphate [75]. It can be observed from the findings that Ca and other exchangeable bases (potassium (K), magnesium (Mg) and sodium (Na)) were found to have increased significantly following lablab growth despite being supplemented by superphosphate and/or *Bradyrhizobium japonicum* inoculant. In fact, K and Ca contents recorded the highest mean values and Mg was the second highest mean value where lablab was planted alone compared with other treatments which accounted for 199.97, 331.45, 104.17 and 38.06 mg/kg greater in K, Ca, Mg and Na respectively than in pre-lablab. The high content of soil exchangeable bases can be attributed to the pH range recorded in soils planted with lablab (5.49–5.81) which is within the ideal range for nutrient availability (K, Ca, Mg and Na) [73]. Additionally, the observed exchangeable bases in the rhizosphere of lablab were most likely related to their supply by mass flow in a rate exceeding that of plant uptake [76]. The significant increase in the exchangeable bases could also be due to the rhizosphere depositions, which aid in making nutrients available for plant uptake [77]. Similar effects were observed in a field trial carried out by [78], who reported a significant increase of exchangeable bases in the soil grown with soybean (*Glycine max*) amended with *rhizobium* inoculant and phosphorus. In addition to the exchangeable bases increase, lablab significantly increased the soil pH in all the treatments in opposition to pre-lablab which was in agreement with the findings of [79], who reported an increase in the cowpea (*Vigna unguiculata*) rhizosphere pH of sandy textured compared to the bulk soil at 45 days after sowing in pot experiment. The pH shift near the root surface of lablab could therefore be vital for nutrient availability. The nutritional shift observed in the current study as a result of lablab growth contrasts with the cereal study which reported a decrease in the soil nutrient concentration near the root surface following their growth and this proves an essential role of the lablab root-induced nutrient improvement for non-leguminous crops to cope with nutrient poor soils [80].

### 4.4. The Effect of Lablab Growth on Soil Micronutrient

Unlike macronutrients, micronutrients are required by plants in smaller quantities, but their remarkable significance is unavoidable since plants depend greatly on them for various plant activities such as photosynthesis, protein synthesis, nodulation in legumes, growth and yield [81,82]. Because of the small quantities of micronutrients required for growth, micronutrients are harmful when applied in larger quantities [83]. Both micronutrient deficiency and/or excess (toxicity) causes impaired growth, which can sometimes lead to crop failure [84]. Since plants acquire micronutrients from the soil, their growth and development play a vital role of returning the micronutrients into the soil [44,85].

In the present study, lablab growth has shown to have a tremendous effect on the soil micronutrient status and has also proven to be conducive even in the absence of

phosphorus fertilizer or *Bradyrhizobium* inoculant since there was no clear difference on soil micronutrients among the treatments but significantly different when compared to the pre lablab soil with the exception of a few micronutrients.

According to [86], Iron (Fe) and Manganese (Mn) contents in the soil pre-lablab growth were below the critical range (<25 mg/kg Fe and <60 mg/kg Mn) which may indicate a deficiency. Based on the initial content of Fe and Mn from the sampling site, the inadequate level of Fe (0.3 mg/kg) and Mn (31.1 mg/kg) could be attributed to the sandy nature of the soil. Sandy soils are generally oxygenated due to their larger pore spaces and under such conditions, the reducing potential of Fe and Mn is hindered and thus likely to limit their availability [87]. Fe deficiency may result in stunted plant growth and low quality in plant yield [88]. On the contrary, Mn deficiency has a considerable effect on photosynthesis, but observable symptoms of chlorosis, appear only when plant growth and yield are severely impeded [82]. The results of the study show that although Fe and Mn contents were still below the critical level, it is interesting to note that the significance increases were tremendous. Fe content was on average 21 mg/kg greater in lablab planted alone (21.3 mg/kg) compared to the pre lablab (0.3 mg/kg) while Mn manganese increased significantly by 14.5 mg/kg in lablab soil relative to the pre lablab. Ref. [89], discovered a slight increase in the Fe content post lablab growth under field conditions, which is consistent the current study. Studies on other legumes such as mung bean (*Vigna radiata*), moth bean (*Vigna Aconitifolia*) and cluster bean (*Cyamopsis tetragonoloba*) on soil micronutrients have found the increased levels of Fe and Mn in the rhizosphere as compared to the non-rhizosphere soil [84].

The study also revealed that soil extractable micronutrients copper (Cu), boron (B) and chloride (Cl) were significantly improved by lablab growth as compared to pre lablab, although zinc (Zn) content was not influenced. The significant increase in the content of these micronutrients could have been attributed to the low-molecular-weight compounds such as amino acids exuded by the roots which aid in the mobilization of the micronutrients and thus making them available for plant uptake [90,91]. The micronutrients can also be influenced indirectly through the microbial activity in the rhizosphere induced by nodulation particularly in legumes. For instance, Ref. [92], associated increased Mn availability in the soil with Mn-reducing bacteria (*Pseudomonas* sp.) which produce chelating agents that form soluble complexes with $Mn^{2+}$. Nonetheless, micronutrient availability in the soil planted of legumes is influenced by the combined impact of soil properties (e.g., soil pH), plant roots and microbes present in the rhizosphere [93,94].

## 5. Conclusions

Total N did not differ among the superphosphate and *Bradyrhizobium japonicum* inoculant treatments as a result of lablab growth, however, planting lablab without the treatments increased the total N significantly. This study suggests that this legume crop has a potential to be effective in contributing total N into the soil without being supplemented by inorganic phosphorus fertilizer or *Bradyrhizobium japonicum* inoculant. Subsistence farmers, such as those in rural village without sufficient access to inorganic and/or synthetic P fertilizers may rely on lablab for enhancing N content of the soil following the harvest of the usually grown non-leguminous crop such as maize. Lablab has also shown to be effective in balancing and fixing the N content in the soil since it significantly increased $NO_3^-$ that was initially low and decreased $NH_4^+$ that was initially high on initially soil sampled and analyzed pre-lablab growth.

Apart from Zn, lablab growth has shown to have a tremendous effect on the soil micronutrient status (Fe, Mn, Cu, B and Cl). This leguminous crop has also proven to be conducive even in the absence of phosphorus fertilizer or *Bradyrhizobium* inoculant, as there was no clear difference in terms of soil Fe, Cu, B and Cl content among the treatments but significantly different when compared to pre-lablab soil. Based on the results of the present study, Mn soil content was significantly increased when superphosphate was applied at 60 and 75 kg/ha as well as when lablab seeds were treated with *Bradyrhizobium* inoculant. This

offers farmers the option of supplementing lablab's potential to increase Mn content with either phosphorus fertilizer or *Bradyrhizobium inoculant* of a moderately acidic Savanna soil. Both Local farmers and emerging farmers who struggle to improve the depleted and low fertility status of their soils, particularly Savana soils may cultivate lablab in replacement of chemical fertilizers which are normally highly cost effective.

**Author Contributions:** Conceptualization, L.M.M. and L.E.N.; methodology, L.E.N.; software, L.M.M.; validation, L.M.M., G.M.M. and L.E.N.; investigation, L.M.M.; resources, L.M.M.; data curation, L.M.M.; writing—original draft preparation, L.E.N.; writing—review and editing, L.E.N. and G.M.M.; visualization, L.E.N.; supervision, G.M.M. and L.E.N.; project administration, L.E.N.; funding acquisition, L.M.M. and L.E.N. All authors have read and agreed to the published version of the manuscript.

**Funding:** This research was funded by National Research Foundation, grant number PR_SFH220123657392 and Unisa M and D bursary.

**Data Availability Statement:** Data supporting will be provided upon request when under review.

**Acknowledgments:** The authors would love to thank Motshepiri village's chief and tribal community for allowing us to collect their soils for experimental use. Additionally, the authors would love to thank University of South Africa for granting us permission to use the greenhouse for conduction of the experiment. The authors would love to thank for the provision of greenhouse where the study was conducted, (Unisa) M and D bursary for funding the entire project.

**Conflicts of Interest:** The authors declare no conflict of interest. The funders had no role in the design of the study; in the collection, analyses, or interpretation of data; in the writing of the manuscript; or in the decision to publish the results.

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
