# Peer review of "The Influence of Lablab Purpureus Growth on Nitrogen Availability and Mineral Composition Concentration in Nutrient Poor Savanna Soils"

_agronomy, doi:10.3390/agronomy13030622_

Round 1
Reviewer 1 Report
MS: agronomy-2064571
Title: The influence of lablab growth on the nitrogen availability and mineral composition on nutrient poor savanna soils
Authors: Miranda Latoya Mthimunye, Gudani Millicent Managa, and Lufuno Ethel Nemadodzi
This study examines the effects of lablab growth on soil fertility, both with and without amendments. It is a straightforward comparison, and the changes are significant. It is clearly a beneficial practice, especially for low-income, rural farmers working with low-nutrient soils. Thus, the study provides valuable information. However, substantial revision is needed before it is publishable. One of the biggest concerns is the lack of methods description. How long did the study last? How was the weather maintained in the greenhouse? Additionally, the control in the study is not a true control. The comments below are separated into major and minor comments, with the line numbers identified.
Major Comments
Lines 65-78, 106-108: Lablab has many favorable characteristics. Please elaborate more on the significance of using Lablab over other legumes.
Lines 142-162: Please elaborate on how the climate was maintained in the greenhouse, especially in terms of temperature and humidity. Were these manipulated similarly to typical weather patterns for the region?
Lines 151-154: Why were these amendment choices made? What were their significance? Please elaborate with one-to-two sentences each.
Lines 151-154: Can the soil planted with lablab and no amendments be considered a control? Wouldn’t the control be just the original soil under the same weather? Undoubtedly, there was an effect from planting the lablab, but without a true control, it is difficult to judge the magnitude of the effect.
Line 162: How long did the experiment run? How was weather maintained? Please provide more specific for the methods.
Lines 468-470: Why is there no increase in total N under the superphosphate and Bradyrhizobium japonicum inoculant treatments, eve with the lablab growth?
Lines 472-475: This might actually be a good way to end the abstract. This statement highlights the importance of the study. However, it is still unclear why lablab is better than other legumes.
Minor Comments
Line 2: The word “lablab” could be considered a typo. Using the full genus-species name (Lablab purpureus) would eliminate any confusion. The revised title would be the following: The influence of Lablab purpureus growth on nitrogen availability and mineral composition in nutrient poor savanna soils”. The same comment applies for the abstract.
Line 7: Please remove “levels”.
Line 9: Please replace “as” with “being”.
Lines 9-11: This sentence can be confusing to some. Perhaps dividing it into 2 sentences may help. Is the following suggestion correct?
“Randomized Complete Block Design with the following four treatments: (1) control; (2) Bradyrhizobium japonicum inoculant; (3)superphosphate; and (4) Bradyrhizobium japonicum inoculant + superphosphate. The superphosphate was applied at three different levels (45, 60 and 75 kg/ha).”
Lines 16-17: Please move “with the exception of Zn” to the end of the sentence.
Lines 22-28: This section must be deleted.
Line 31: Please replace “provides” with “providing”.
Line 32: Please remove “For many years”.
Line 35: Please replace “has” with “have”.
Line 36: Please remove “recognized and”.
Line 39: Please replace “gets” with “is” and remove the comma after “[1],”.
Line 40: Please replace “litters” with “litter”.
Lines 47- 52: These sentences can be removed as they contain basic knowledge that do not directly relate to the study,
Line 54: Please remove “mainly accounting”.
Line 56: Please remove the comma after “[12]”.
Line 58: Please remove “N form”.
Line 76: Please remove the comma after “[21]”.
Lines 80-82: The sentence seems awkward mentioning both similarities and contrasts, as well as other topics.
Line 97: Is “lablab” considered as plural? My thought is it would be single.
Line 98: Please replace the semicolon with a period.
Lines 98-105: Please move these sentences before line 94. Then add in its place another sentence or two tying it back to the material in lines 54-64.
Lines 114-115: Is “World weather online, 2022)” supposed to be a reference? Why is there only 1 part of the parentheses?
Lines 116-117: The material about soils and intercropping should be in separate sentences.
Lines 115-118: There is too much information in this sentence which makes it quite confusing.
Line 125: Please remove “foreign materials such as”.
Lines 128-129: Please rename this section as “Pre-planting soil physical and chemical analysis” and then delete the word “Pre-planting” from the beginning of the sentence.
Line 144: Please replace “ranges from” with “of”.
Line 175: Where is figure 1?
Lines 200-205: Wouldn’t one expect the Phosphorus to increase after adding phosphorus fertilizer? Am I missing something here? What is the significance of this result?
Line 268: Please replace “significance” with “significant”.
Line 314: Please rewrite as “ In the pre-lablab growth soil, the total nitrogen…”
Lines 318-318, 354-356: Why is important to know about the effects of the continuous maize production on the collected soil?
Line 323: Please remove the comma after “[50]”.
Line 328: What is “Nous”?
Line 355: Is this the correct heading? It is repetitive.
Line 360: What is “chapter 4”?
Author Response
Beside the initial favourable characteristics offered by Lapla purpureus, we indicated its leaves and pods are widely consumed in the Northern part of South Africa, which makes it easily accessible. Therefore, local farmers can grow it as a means to improve the status of their soil particularly soils with depleted. nutrients.
Line 151-154: soil planted with lablab without amendments had to be used as control to achieve the main aim of the study which is well articulated on the abstract. Amendments were added as a means to observe any change (significant/ least significance/ non-significant) differences in the initial macro and micronutrients pre-lablab growth.
Total N only showed a significant increase where lablab was grown and a non-significant where lablab was cultivated with Bradyrhizobium inoculant and superphosphate. These findings have opened up a future study to investigate if the different higher levels of the above mentioned could have an impact on the total N.
All the minor comments have been incorporated as per the reviewers' suggestions. Previous errors were replaced and now in order. Kindly note that the effect/impact was on micronutrients and macronutrients therefore not repetitive.

Reviewer 2 Report
This study reported the impacts of lablab growth on soil macro and micro nutrients on a poor savanna soils, based on the results of a greenhouse pot experiment that lablab was planted with or without P fertilization and inoculation. In general, the experiment was well designed and the paper is well organized. The conclusions are meaningful in guiding the local farmers to intercrop lablab with other crops. However, I have some major concerns as follows:
1. This study focused on the lablab growth on soil macro and micro nutrients, but the impacts of legume on soil macro and micro nutrients in previous studies were not reviewed and summarized in the whole introduction. So, review the previous papers about the legume growth on soil nutrients and summarize the results, then hypothesize the potential influences of lablab growth.
2 Explain why the influences of the lablab growth were based the comparison of soil nutrients between rhizosphere soils after lablab growth and bulk soils before the growth, but not between the bulk soils after and before growth? To our knowledge, the benefit of lablab growth to intercropped non-legume crops is mainly in bulk soils. Also, the changes in rhizosphere nutrients could also result from the movement of nutrients from bulk to rhizosphere soils.
3. It is confusing that rhizobium inoculation and P fertilization did not elevate the positive impacts of lablab growth on soil N. The authors need discuss these results and explain why.
4.Some tables are suggested to be replaced by figures, which are more clear.
Author Response
- Although several studies have been conducted on legumes, few of them address their impact on nutrients depletion. However, in the current paper, our findings are supported by study conducted on peas and cow peas (Line 321 and 323). we are hoping that this article will play a huge role in awareness on the use of legumes as a measure to improve the nutrients status of the soil.
- In this study, lablab was not cultivated as an intercrop but a monocrop and a means to remedy nutrients depletion due to continuous cultivation of maize plants. Micro and micronutrients were analysed pre and post lablab growth. The differences between the pre-lablab growth versus post-lablab growth had to be analysed to achieve the main aim of the study which was explained through the findings of the study.
- The non-significant increase on total N has opened a gap for future studies which will investigate the impact of higher rates /concentration levels of Bradyrhizobium inoculant and superphosphate on total N.

Reviewer 3 Report
Planting of legumes could improve soil fertility. In this paper authors used pot experiment to study influence of lablab growth on the nitrogen availability and mineral nutrient content of poor savanna soils which will be helpful for readers to understand the reason that legume could improve soil fertility. But the present datas in this paper can not reveal the reason. The author needs to supplement datas of nodule number and biomass to explain why soil fertility was improved. Datas in this paper is insufficient . The novelty of the data presented in the paper is unclear.
Some other issues.
1. The title need to be revised. Mineral composition changed to “Mineral composition concentration”. Change “on” into “of”.
2. The systematic abstract is missing. Introduce the need for study in 1-2 lines. Then please give a clear-cut point problem source as a problem statement that is tackled in the current study. Also, give a logical reason for selecting the current strategy or treatments. Then provide a definitive conclusion withdrawn through the research.
3. In Key words, nitrogen forms; total nitrogen
4. Give a logical reason for selecting the current strategy and the novelty of this research in introduction should be clarified. Add the progress of legumes in improving soil fertility in introduction.
5. The authors mentioned that a two-way analysis of variance(L165)L was used in statistical analysis, but the expressions in table1、table2 and table3 are not correct. The interaction of superphosphate and inoculant didn’t show.
6. Line 117 what kind of soil used in this experiment according to American Soil Classification?
7. The form of tables is not standardized and does not meet the requirements of the journal.
8. In table3 the units of Cu and Zn are not correct.
9. Except superphosphate whether the other nutrient such as N and K was applied in this pot experiment? The nutrient content in pre-lablab soil is very low. Why does potassium content in post lablab increase dramatically?(table 2)
10. Languages in this paper need to be improved. Some sentences need to be rewritten.
Such as Line52-56,L126,L160 and L355-356.
11. Change”Nitrogen”into “nitrogen”
12. Add”contents”behind”NO3-“(L7)
13. Change” Highworth”into “highworth”.(L79)
14. Change” N”into “N2”.(L97)
15. Maximum air temperature in greenhouse is44.9 °C. If it will have temperature stress for plants?
Author Response
- The title has been changed per your suggestion.
- Adding more words would make the words in the abstract beyond the journal's required number of words which is 250. however, the aim is well articulated in the introduction.
- According to the author's guidelines, the keywords should not form part of the title. nitrogen forms and total nitrogen does not form part of the title.
- The novelty of the study has been added as per your suggestions (this is the very first study conducted in South Africa using Lablab as a means to tackle low soil fertility. This is going to play a huge role in farmers as they wouldn't have to depend on inorganic and or chemical fertilisers which are very costly. Few study have been conducted on the use of legumes towards improving soil fertility. However, line 321 and 323 mentioned a few of the findings.
- The initial error was corrected and replaced with one-way ANOVA, refer to line 171
6. The soil used in the experiment is Savanna soils are classified as Alfisols and Ultisols in American soil classification
8. The values were corrected, refer to L283 and 291.
9. The study only reported the results as they are, the cause of drastic increase in potassium post lablab growth is identified as a gap to be investigated for future studies in the near future.
10. Sentences rephrased
11-14. All the suggestions have been incomporated.
15. Plants were never subjected to the maximum temperature; it was only mentioned in this article for the benefit of total comprehension of the greenhouse condition.

Reviewer 4 Report
Overall, the topic of the manuscript was interesting and soundness, however, some of the parts need to improved to make it more clear.
1. You should mention the lablab were cultivated in each treatments in the Abstract (L 8-11)
2. “… proved to have a significant impact on both the 16 soil macro (P, K, Ca, Mg, and Na) and micronutrient level (Fe, Mn, Cu, B and Cl).” You should express it clearly that what impact the macro and micronutrient level, and how does it affect? (L 14-17)
3. It is better that the significance of this study was introduced in the end of the Abstract.
4. “0 How to Use This Template” should be removed from the manuscript. (L 22-28)
5. In the Introduction part, you mentioned the background of the study, the nutrient of nitrogen, and wrote a lot of information about the Lablab. In my point of view, the properties and usages of the Lablab should be simplified. What is more important, the purpose of the treatments setup should be introduced. Why did you use Bradyrhizobium japonicum inoculant, and why did you set three levels of superphosphate? Are there any literature support? You did tell us.
6. You should clarify the purpose of the experiment further and respond to the topic at the end of the Introduction.
7. “where lablab was planted without amendment by Bradyrhizobium japon-178 icum inoculant and/or superphosphate relative to soil pre-lablab” should be removed into 2. Materials and Methods. (L 178-179)
8. “Macronutrient content” was not appropriate in the first line of Table 2, as CEC and pH were not Macronutrient contents.
9. Any results of lablab growth status of different treatments?
10. Are there and connections of soil nitrogen content and soil macronutrient or micronutrient?
11. The heading of ”4.2 The effect of lablab growth on soil total nitrogen contents” need to be corrected.
12. The Conclusions of the manuscript should be more concise and enlightening.

Author Response
- Mentioned, refer to line 13
- The maximum words for abstract as per the author's guidelines is 250, adding more words would have meant that word will be above the specified number.
- Significance is well indicated in line 21-=23.
- Removed.
- Suggestions incorporated in line 110-133.
- Suggestions incorporated in line 130-133.
- Line 178-179 is reporting on the results of the study.
- Table 2 is reporting on the macronutrients, CEC and soil pH, hence the commas in between.
- Yes, Due to the nature of the study, two more papers were prepared to report the findings, with the response of lablab growth to different treatments being the second paper. However, the results cannot be shared as the manuscript is currently under review.
- No, there are no connection between soil nitrogen content and macro/micronutrients.
- Heading on 4.2 changed, refer to line 366.
- Conclusion added, refer to line 520-523.

Round 2
Reviewer 1 Report
MS#: agronomy-2064571
Title: The influence of lablab purpureus growth on nitrogen availability and mineral composition concentration in nutrient poor savanna soils
Authors: M.L. Mthimunye, G.M. Managa, L.E. Nemadodzi
Comments: In this study, the authors provide the benefits of the lablab purpureus in helping savanna soils improve their fertility. This is an important finding and worthy of publishing. However, this paper needs another round of moderate revision. The introduction contains background information that is not germane to the essence of the study. It is also jumpy at times. Some sentences seem out of place.
Some clarification is also needed with the methods and especially the climate of the greenhouse. Below there are some editorial suggestions followed by questions and comments that must be addressed before the paper is published.
Editing suggestions:
Line 8: Please replace “have” with “has”.
Line 9: Please move the sentence “Soil used in this pot experiment was obtained from Motshephiri village which has low total N, low NO3- and high NH4.” (in Lines 12-14) between the two sentences in Line 9.
Line 15: A suggestion for “soil pre-lablab” is “original soil”.
Line 71: Please replace “tolerance” with “tolerant”.
Line 72: Please remove “during”.
Lines 73-75 & Lines 83-84: These seem repetitive.
Line 77: Pretty much everything up until this point is not needed for the purposes of this paper. The paper should start here. The very first sentence (lines 35-37) can be kept.
Lines 78-81: Please move this sentence after the following sentence.
Lines 98-102 & 109-115: These sentences are not needed.
Line 163: Please remove “Core”. This sampling does not sound like taking a core.
Line 202: Please replace “was” with “were”.
Line 212: Please replace the second “using” with “with the”.
Line 221: Please replace “were” with “was”.
Line 223: Should “along” be “between”?
Line 238: Should there be a period between “treatments” and “The”.
Line 249: The parentheses should be around “no significant difference”.
Line 260: Please remove the word “below”.
Line 267: What does the asterisk mean?
Questions & Concerns:
Line 164: How much downcore variation in micro- and macro-nutrients concentrations do you have in the top 30 cm? Do the concentrations have a non-linear profile? How well does the rooting depth match this 30 cm sample depth?
Lines 167 & 184: Why were the pre-planting soils sieved at 4 mm and the post-planting soils sieved at 2 mm? Could this difference have affected your results? Perhaps with a concentrating effect by removing coarser, “less-active” soils?
Line 192: If the plastic pots were separated, please explain the benefits of using the Randomized Complete Block Design.
Line 196: This is not a true control. A true control would experience the same climate conditions but have nothing planted. Is it possible to come up with a different name?
Line 203: If you watered the pots with 250 ml of water every second day for six months (i.e. 90 days), then how does this total volume of 22.5 L correspond to the annual rainfall of Roodepoort, Johannesburg? This suggests 884 mm of rainfall during the experiment, which is higher than the annual average of 604 mm.
Lines 159, 187-188; 209-209: How was the weather regulated to match that of Roodepoort, Johannesburg? I am a little confused with the annual temperatures ranging from 8 to 28 degrees C and the pot experiment being conducted in the greenhouse with a minimum and maximum air temperature range of 7.4 and 44.9 degrees C.
Author Response
Line 8: have was replaced with has.
Line 9: line 12-14 moved to line 9.
Line 15: thank you for the suggestion, original soil is added "with herein referred to as pre-lablab''. We believe that the word pre-lablab will make it easier for the readers when comparing the results with post lab-lab growth.
Line 71: tolerance replaced with tolerant.
Line 72: during removed.
Line 83-85 which was repetitive of line 73-75 has been removed.
Line 77: Thank you very much for your suggestion, however, the very sentence was included in the paper as per suggestions from the other reviewers. Removing would mean contradiction with what the other reviewers recommended. we hope you will find our response in order.
Line 78-81: same response as provided for line 77.
Line 98-102: Thank you very much for your comment. We (the authors) believe that since the study was conducted in South Africa where the popularity of Lablab is only common among the few marginalised tribes, bringing the common names and other countries where it is widely consumed is of paramount important to bring awareness about Lablab to the readers (in and out of academia), researchers and other scientist.
Line105-109: these sentences were added per recommendations made by other reviewers, removing them would mean contradicting their suggestions.
Line 163: Core removed.
Line 202: was replaced with were.
Line 212: second using replaced with "with the ".
Line 221: were replaced with was.
Line 223: along replaced with between.
Line 238: Period added.
Line 249: Parentheses added.
Line 260: below removed.
Line 267: Thank you for the question, however, the word asterisk does not seem to appear on the manuscript. We would like to believe that it might have been a typo which has since been corrected.
Questions and concerns
Line 164: Macro and micro-nutrient concentrations within 30cm soil depth vary depending on numerus edaphic factors such as biological activity, land use, extent of disturbance, environmental conditions and particles size distribution. In our study, macro and micro-nutrient concentrations varied extensively which might be attributed to long-term disturbance of the soil through tillage practices.
The soil used as medium growth was initially used to grow maize, which has a rooting depth of 20-38cm. Kindly note that the effective root depth of maize was considered but not that of lablab since the soil had previously been subjected to nutrient depleted as shown on table 1, 2, 3 (pre-lablab).
Line 167-168 4mm sieve was used on the original soil to remove stones, weeds, debris and to break the clumps in preparation of the experiment. 2mm sieve was used in preparation of chemical analysis (prior and post lablab growth), in order to obtain the small soil particles where the nutrients are adhered to as opposed to the coarser particles as prescribed by Kettler et al., 2001. Please note that the citation has been added on the manuscript (*see line 88), the citation was also added on the reference list in 56.
Line 192: the plastic pots were all in one space in the greenhouse. Each plastic pot had its own lablab grown, RCBD was used as a means of blocking any influence from treatment 1 /2/3 as advised by the institution residence statistician. kindly note that the proposal was presented to the department of Agriculture and Environmental health where academics scrutinised and gave their inputs before the commencement of the project and RCBD was approved.
Line 196: Thank you for your question, control in our study was referred to original soil where lablab (herein referred to as Pre-lablab growth) was grown without application of different superphosphate rates nor Bradyrhizobium inoculant. unfortunately, there is no other word we can use in this regard. we hope you find our response in order.
Line 203: The plastic pots were designed to have holes underneath/ at the bottom to allow drainage of excess water and proper aeration as well as to prevent water saturation. Tyhe amount of water used was in alignment with the level of drainage observed.
Line 159, 187-188, 209: Thank you very much for your question. Kindly note that the greenhouse temperature was set to match that of Motshepiri Village. 7-44 degrees Celsius was mentioned to provide a total understanding of the maximum it can reach which was not the case in our study hence the word ''ranges'' was used. We trust that you will find our response in order.
Reviewer 2 Report
An important question in this study is that why the soil nutrients were compared between the rhizosphere soils after lablab growth and bulk soils before lablab growth, but not compared between both the bulk soils after and before lablab growth. It is well known that bulk and rhizosphere soils differed greatly in nutrients. I think it is necessary to explain this experiment design to verify the reasonability of the experiment design.
Author Response
Please note that this was a pot-experiment (18cm) as indicated in line 188 of the manuscript (Experimental design and layout) which was conducted in the greenhouse. The roots of lablab had extended deep and wide in the pot. Bulk soil used in the study served as control (pre-lablab growth). The rhizosphere soil in the study was the post-lablab growth. At termination, the lablab crops were excavated and soil samples from the pots were taken for analysis. It wouldn't have made sense to collect bulk soil post-lablab because this would deviated from the aim of the study, which was to'' determine the effect of lablab growth on the soil depleted of nutrients (macro and micro)''. From the findings of the study, it is concluded that farmers can grow lablab as measures to improve the nutrients status of the soil. After harvesting lablab, the soil nutrients status would have improved.
Reviewer 3 Report
The quality of the manuscript was improved after revision. But there are some problems needs to be revised furrher.
1) There are two factors in this experiment including P level and inoculation .So the author should use the two-factor method to do ANOVA analysis instead of single factor. Then interaction between P and inoculation could be analyzed. ANOVA analysis of table 1-3 should redo.
2) “nitrogen source”should be changed to “P levels” In table 1.
3)line 15 add“concentration” behind"NH4+”
Author Response
1. Thank you for your comment, we apologise for an error or misunderstanding that might have resulted in the way we presented our tables. The experiments/ treatments were ran separately and hence no need for an interaction as Japonicum remained constant throughout, hence one way analysis. The tables were amended for easy reference and understanding.
2. Table 1 only focus on Nitrogen sources / forms (Total Nitrogen, Nitrate and ammonium) which did not change to phosphorus (P). However, the growth of lablab only changed the initial state of Total N, Nitrate and ammonium. Please note that Table 2 focused on Phosphorus (P) among other macronutrients.
3. Added
Reviewer 4 Report
1 The significance of this study was introduced. However, some of the contents are repeated in the last two sentences of the Abstract. It is suggested to merge the two sentences.
2 It is suggested that incorporate the third and fourth paragraphs of the Introduction, because some of the information is repeated.
3 The last paragraph of the Introduction should be improved. The objectives of this study should be introduced here.
Author Response
- Sentences merged into one.
- Repeated sentences removed.
- Introduction improved and objectives added (*see line 148-153).